# Hidradenitis Suppurativa in Association with Ulcerative Proctitis: Surgical Management in a Refractory Case to Topical and Systemic Treatment

**DOI:** 10.3390/reports7010013

**Published:** 2024-02-17

**Authors:** Ioana-Valentina Stoenică, Mihai Cristian Dumitrașcu, Aida Petca, Răzvan-Cosmin Petca, Florica Șandru

**Affiliations:** 1Dermatology Department, Elias University Emergency Hospital, 011461 Bucharest, Romania; ioana-valentina.stoenica@rez.umfcd.ro (I.-V.S.); florica.sandru@umfcd.ro (F.Ș.); 2Department of Dermatovenerology, “Carol Davila” University of Medicine and Pharmacy, 020021 Bucharest, Romania; 3Department of Obstetrics and Gynecology, “Carol Davila” University of Medicine and Pharmacy, 020021 Bucharest, Romania; mihai.dumitrascu@umfcd.ro (M.C.D.); aida.petca@umfcd.ro (A.P.); 4Department of Obstetrics and Gynecology, University Emergency Hospital of Bucharest, 050098 Bucharest, Romania; 5Department of Obstetrics and Gynecology, Elias University Emergency Hospital, 011461 Bucharest, Romania; 6Department of Urology, “Carol Davila” University of Medicine and Pharmacy, 020021 Bucharest, Romania; 7Department of Urology, “Professor Doctor Theodor Burghele” Clinical Hospital, 061344 Bucharest, Romania

**Keywords:** hidradenitis suppurativa, ulcerative proctitis, surgical treatment, unroofing, punch debridement

## Abstract

Hidradenitis suppurativa (HS), a challenging dermatological condition, can be described as a chronic, painful, follicular, occlusive disease that is characterized by painful nodules, abscesses, and sinus tracts generally located in the intertriginous skin areas. This disorder places a considerable burden on affected individuals and often leads to significant impairment in quality of life for those grappling with its persistent and recurrent nature. We present the case of a 20-year-old male patient known to have ulcerative proctitis and diagnosed with hidradenitis suppurativa in 2020, for which he underwent various topical and systemic treatments, with periods of remission and recurrent flares without managing to obtain complete remission of HS. In our dermatology service, the patient underwent two surgical unroofing procedures combined with punch debridement, with the wounds being allowed to heal by secondary intention. This choice of treatment delivered great results with favorable outcomes, without recurrence of the lesions, proving to be an effective method for managing HS. By presenting this case, we underline the role of surgical procedures in managing HS, and our desire is to emphasize the importance of comprehensive understanding of this enigmatic and complex condition for more effective management strategies in patients with refractory disease.

## 1. Introduction

Hidradenitis suppurativa, also known as acne inversa [1], stands as a dermatological enigma characterized by the chronic and debilitating nature of its inflammatory skin lesions. HS is a chronic follicular occlusive disease affecting the follicular portion of the folliculopilosebaceous units (FPSUs), characterized by the development of painful nodules, abscesses, and sinus tracts, that places a considerable burden on affected individuals [2]. This disorder primarily affects the intertriginous areas, such as the axillae, groin, and buttocks, and often leads to significant impairment in quality of life for those grappling with its persistent and recurrent nature [3].

The epidemiology of HS is a dynamic landscape, with estimates suggesting a prevalence of approximately 1% in the general population [4]. However, this prevalence might be underestimated due to underdiagnosis and misdiagnosis, underscoring the need for heightened clinical awareness. HS typically emerges in early adulthood, presenting a peak onset in the second and third decades of life. The gender predilection is notable, with a higher prevalence in females, although the disease’s severity may be more pronounced in males [5].

The pathogenesis of HS involves a multifaceted interplay of genetic, immunological, and environmental factors. Current evidence suggests a polygenic inheritance, with mutations in genes associated with inflammation, innate immunity, and the follicular epithelium contributing to disease susceptibility. Dysregulation of the immune response, particularly involving the interleukin pathway, plays a pivotal role in the inflammatory cascades observed in HS [6].

The probable instigating factor for the initial formation of HS lesions is follicular occlusion, which stems from the proliferation of ductal keratinocytes, resulting in follicular hyperkeratosis and blockage. It is suggested that associated factors within this process are the hormones and nicotine, which have a negative impact on the follicular epithelium. The resulting anoxia within the follicular duct, induced by follicular epithelial hyperplasia, contributes to the disturbance of the usual terminal differentiation of follicular keratinocytes, obstructing the follicular passage and subsequent follicular rupture [7]. 

The occlusion of hair follicles, subsequent inflammation, and bacterial colonization contribute to the formation of abscesses and sinus tracts characteristic of HS lesions. Skin tunnels that have been firmly established ultimately emerge onto the surface of the skin, undergoing persistent inflammation. Lifestyle factors such as smoking, obesity, and poor diet further exacerbate the disease, highlighting the importance of a comprehensive understanding of the disease’s pathophysiology [8]. Furthermore, several studies indicate a correlation between HS and inflammatory bowel disease (IBD) [9], both Crohn’s disease (CD) and ulcerative colitis (UC) [10], as they exhibit some similarities to HS regarding clinical manifestations, genetic predisposition, and immunological characteristics.

Regarding its clinical manifestation, hidradenitis suppurativa unfolds with painful, inflamed nodules that may progress to abscesses and sinus tracts. These lesions predominantly manifest in intertriginous areas, including the axillae (most common site), inguinal area, inframammary, and perianal regions. The formation of skin tunnels, clusters of exposed comedones (referred to as tombstone comedones), and scarring are outcomes observed in cases of recurring or persistent disease. The disease course is generally marked by recurrent flares and remission periods, which contribute to its chronic nature [11]. 

Disease onset typically occurs post-puberty, often in the second and third decades of life, and there is a correlation between the earlier onset of HS and a more extensive and widespread disease [12]. Furthermore, diagnosis of the condition is often delayed, especially in cases categorized as ‘mild’ and intermittent, the primary inflammatory nodules being commonly misdiagnosed as furuncles [13]. On top of that, the lack of a specific diagnostic test demands a meticulous clinical assessment for the diagnosis of hidradenitis suppurativa. The diagnosis relies on the presence of typical lesions, typical topography, and chronicity and recurrences [2]. Diagnostic criteria, including the Hurley staging system and the International Hidradenitis Suppurativa Severity Score System (IHS4), aid in categorizing disease severity and guiding treatment decisions. 

The Hurley clinical staging system is commonly employed to categorize patients with HS into three levels of disease severity: Stage I: formation of abscesses (single or multiple) without skin tunnels or cicatrization/scarring; Stage II: recurrent abscesses accompanied by skin tunnels and scarring, with single or multiple lesions widely separated; and Stage III: widespread or nearly widespread involvement, or the presence of multiple interconnected skin tunnels and abscesses across the entire affected area [14]. The majority of HS patients typically fall under stage I disease [15].

IHS4 represents a validated instrument for dynamically evaluating the severity of hidradenitis suppurativa. IHS4 is determined by tallying the number of nodules (multiplied by 1), abscesses (multiplied by 2), and draining tunnels (multiplied by 4). A cumulative score of 3 or below indicates mild disease, 4–10 denotes moderate severity, and 11 or higher indicates severe HS [16]. By incorporating these criteria, IHS4 offers clinicians a standardized framework for assessing and monitoring HS severity, facilitating improved treatment decisions and patient management strategies.

Managing hidradenitis suppurativa necessitates a nuanced, multimodal approach, reflecting the chronic and relapsing nature of the disease. Lifestyle modifications, encompassing weight loss and smoking cessation, emerge as foundational pillars of disease management [17]. The severity of the disease has a significant impact on the approach to treatment. 

In milder cases (Hurley stage I), therapy aims to reduce the burden of the disease (i.e., to limit the development of lesions and inhibit the progression of the disease) and to improve acute, symptomatic lesions. The treatment consists of topical and systemic antibiotics that target bacterial colonization, mitigating inflammation [18]. Although the initial therapy in these cases consists generally of oral tetracyclines, topical clindamycin is occasionally considered and might serve as a viable initial approach for individuals with mild disease. Oral antiandrogenic medications and metformin represent supplementary treatment choices that can be employed independently or in combination with antibiotics. Further measures for managing symptomatic inflamed nodules encompass intralesional corticosteroid injections, punch debridement (partial unroofing), and the application of topical resorcinol [19].

A spectrum of interventions comes into play for moderate to severe disease (Hurley stage II or III). Oral antibiotic therapy, intralesional corticosteroid injections, systemic immunosuppressants, and biologic therapies—the latter targeting specific inflammatory pathways—showcase efficacy in controlling disease progression. Furthermore, surgical options, including drainage of abscesses, surgical unroofing, and excision of involved tissue, prove beneficial in refractory cases, although necessitating carefully weighing risks and benefits [20].

In this case report we present the case of a 20-year-old male known to have ulcerative proctitis and diagnosed with hidradenitis suppurativa in 2020, for which he underwent two surgical interventions of incision and drainage in another service, various topical treatments (clindamycin 1% solution, resorcinol 15% cream and topical antiseptic washes), and systemic treatments (oral doxycycline), with periods of remission, recurrent flares and persistent worsening of symptoms in recent months. This case report will outline the surgical treatment options chosen, the decisions made, and the subsequent outcome for our patient. By presenting this case, our intention is to underscore the significance of possessing a thorough understanding of this intricate condition. Such understanding is pivotal in devising more effective management strategies, especially for patients grappling with refractory disease.

## 2. Detailed Case Description

We present a case involving a 20-year-old male, a non-smoker with a normal BMI known with ulcerative proctitis and diagnosed with hidradenitis suppurativa in 2020, who sought clinical evaluation and specialized treatment in our clinic. 

In 2015, the patient received a diagnosis of mild ulcerative proctitis and underwent a four-week treatment regimen of topical mesalamine at a daily dose of 1 g, leading to subsequent remission of the disease. Since then, there has been a single episode of disease relapse, effectively managed with topical mesalamine without the necessity for ongoing maintenance therapy.

The onset of HS symptoms occurred three years before. Since then, the patient has undergone a range of treatments: two surgical interventions of incision and drainage in another service, various topical treatments (clindamycin 1% solution, one application per day, resorcinol 15% cream, one application per day, and topical antiseptic washes) and systemic treatments (oral doxycycline 50 mg once daily for three months), with periods of remission and recurrent flares, without managing to obtain the complete remission of the disease.

At the current presentation, the dermatological clinical exam reveals typical lesions of HS located in both of the patient’s axillae. In the left axilla, we observed two skin tunnels, one inflammatory nodule, and a thick, linear, rope-like band of scarring (Figure 1a). On the other hand, in the right axilla, we described one solitary skin tunnel and two deep-seated inflamed nodules with minimal scarring (Figure 1b). Following the extent and severity of lesions, the patient was classified as having Hurley stage II disease and an IHS4 score of 15. 

In instances where the prevention of new lesions proves unsuccessful and medical therapy for established and growing lesions is ineffective, surgery becomes the acknowledged approach for managing hidradenitis suppurativa. Considering this, in the current context of our patient’s presentation, we opted for the surgical management of the axillary lesions.

Initially, we conducted the surgical treatment of lesions localized in the left axilla under localized anesthesia using lidocaine 1% (Figure 2a). We performed the surgical unroofing of two skin tunnels (Figure 2b) and the punch debridement of one inflammatory nodule (Figure 2c). The punch debridement procedure utilized a 5 mm circular punch instrument to excise the isolated inflammatory nodule, along with a minimal amount of surrounding tissue. 

In the context of the surgical unroofing technique, the procedure commenced with the delineation of the affected area followed by local anesthesia administration (adrenaline was utilized to prolong the duration of action). Due to the absence of a discernible sinus tract opening, a surgical entry point was established through the overlying skin. A guiding probe was introduced through the surgical entry point and meticulously navigated along the entire length of the sinus tract. Subsequently, an incision was made along the probe’s trajectory, resulting in the detachment of the upper portion of the sinus tract. The entire roof of each tract was meticulously excised, and both the base and margins were explored for concealed entrances to other tunnels. Cellular debris at the base of the tract was debrided, along with any superficial scar tissue and granulation tissue, thereby aiding in the elimination of the biofilm lining the tunnel. Superficial curettage resulted in a uniform, bleeding lesion bed at the subcutaneous level.

After the procedures, the wounds were allowed to heal by secondary intention (Figure 2d), using just some dressing pads with a silicone wound contact layer. Following the intervention, the patient was prescribed a course of antibiotic therapy consisting of Cefuroxime 500 mg, taken twice daily for a duration of 5 days. The patient was given detailed instructions regarding the wound care regimen to be followed at home, which involved gently cleansing the area with an antiseptic solution, followed by the application of a moist dressing containing a neutral ointment, and sealing with a superabsorbent silicon-lined dressing. The patient diligently followed this home care protocol for 2 weeks. After the follow-up assessment, we advised the patient to apply another ointment with antibacterial and scar-reducing properties, containing silver sulfadiazine. 

The patient attended follow-up appointments at two weeks (Figure 3a) and four weeks (Figure 3b) post operation, demonstrating a favorable progression of the lesions with no indication of inflammation or infection.

Subsequently, we performed the surgical management of the lesions localized in the right axilla, performing the surgical unroofing of one skin tunnel and the punch debridement of two inflammatory nodules (Figure 4a), leaving the wounds to heal by secondary intention (Figure 4b). The patient attended follow-up appointments at two weeks (Figure 4c) and four weeks (Figure 4d) post-operation, also showing a favorable evolution. Six months post intervention, the patient is in total remission without any new HS lesions in the axillary region or any other areas. 

## 3. Discussion

Several risk factors have been statistically established as contributing to the development of hidradenitis suppurativa. Factors such as genetic susceptibility, mechanical stresses on the skin, obesity, smoking, dietary habits, and hormonal influences are frequently mentioned as potential contributors linked to the onset or worsening of HS. Our case diverges, as the patient lacked any of these factors mentioned. He was a non-smoker with a normal BMI, maintained a healthy dietary habit, and had no familial history of genetic susceptibility to the condition.

Most notably, our patient was known to have mild ulcerative proctitis, diagnosed in 2015 and, at presentation, in clinical remission after the treatment with topical mesalamine. Several studies indicate a correlation between hidradenitis suppurativa and inflammatory bowel disease [9]. Earlier research suggested that the association between HS and IBD was present only in Crohn’s disease [21], with no apparent link to hidradenitis suppurativa in ulcerative colitis [22]. However, this notion has since been disproved, as subsequent studies are increasingly revealing a correlation between HS and ulcerative colitis as well. The most recent study is that of Bingzhou Bao et al., in 2023; theirs was a bidirectional Mendelian randomization study proving a robust causal relationship between IBD of both subtypes (CD and UC) and HS [23]. Concerning the pathophysiological aspect, there is some evidence indicating that IBD and HS share common clinical manifestations [24], genetic predisposition, and immunological profiles [22]; however, further research is needed to elucidate the pathophysiology of the causal connection between IBD and HS.

As far as treatment is concerned, our patient underwent a variety of interventions to manage the disease. He reported having undergone two surgical procedures involving incision and drainage in another service but experienced a swift recurrence of the lesions. Currently, it has been demonstrated that the customary practice of incision and drainage (I&D) for individual nodules is deemed ineffective and unsuitable for managing hidradenitis suppurativa (HS) [25]. I&D offers only temporary relief, and as it fails to address the actively growing tissue, lesions treated in this manner often experience recurrence [26].

Our patient was classified as having Hurley stage II disease. In this stage of HS, the primary therapeutic strategy to reduce the disease burden involves using oral antibiotics, typically oral tetracyclines or a combination of oral clindamycin and rifampin [27]. Our patient had previously undergone a three-month course of oral doxycycline at a daily dosage of 50 mg, yielding no substantial improvement in the lesions. Typically, doxycycline is prescribed at a recommended dosage of 100 mg, administered once to twice daily [28]. The suboptimal response to the treatment observed in our patient could be attributed to the lower dosage employed during the course. 

On the other hand, for the treatment of acute, symptomatic lesions, additional interventions are necessary that may consist of intralesional corticosteroid injections [29], punch debridement, unroofing [30], and topical resorcinol [31]. For our patient, intermittent use of topical resorcinol 15% cream was employed to address newly inflamed nodules, resulting in temporary alleviation of pain; however, unfortunately, upon discontinuation of the medication there was a rapid recurrence of symptoms. In our case, the various topical and systemic treatments resulted in periods of remission and recurrent flares without obtaining a complete remission of the disease. In this instance, there are alternative medical treatment options to consider, including biologic tumor necrosis factor (TNF) inhibitors like adalimumab [32] and infliximab [33], the interleukin (IL) 17A inhibitor secukinumab [34], and oral acitretin (for patients not of childbearing potential) [27]. However, these treatments were not deemed necessary in our case as the lesions were localized, and the patient was considered suitable for surgical management of the disease.

A diverse array of surgical techniques are employed in the treatment of patients with hidradenitis suppurativa. These methods can be customized based on factors such as disease stage, defect size, bacterial contamination level, anatomical location, and the consideration of scar placement and potential for revision. The most-used surgical techniques are punch debridement with a 5 to 7 mm punch biopsy instrument and surgical unroofing, which involves removal of the superficial portion of the lesion [35] followed by closure by secondary intention [36]; these techniques were also used in our case. For more severe cases, extensive excision may be necessary to address a region affected by chronic or widespread hidradenitis suppurativa (Hurley stage III) when conservative medical and surgical interventions prove ineffective [37]. 

In our case, the surgical intervention yielded outstanding results. At the moment, more than six months post surgical treatment, the patient remains free of any recurrence of lesions. This affirms the effectiveness of surgical methods as an impactful approach for managing moderate-stage hidradenitis suppurativa (Hurley stage II). This strategy proves particularly advantageous for patients with inflammatory nodules and skin tunnels that do not respond to traditional topical or systemic therapies.

## 4. Conclusions

Based on a thorough case analysis, the patient under consideration exhibited no apparent common risk factors for developing HS, with the only noteworthy association being the diagnosis of ulcerative proctitis. This correlation underscores the complexity of the disease and highlights the interplay of genetic, immunological, and environmental factors in the development of this disorder. Despite being unresponsive to initial topical and systemic therapy, the patient responded remarkably well to the surgical management of the disease. The present case underlines the importance of surgical procedures in managing moderate-stage hidradenitis suppurativa (Hurley stage II). This comprehensive exploration, spanning the epidemiology, pathogenesis, clinical manifestations, staging, diagnosis, and most of all the surgical treatment of HS, underscores the importance of thoroughly understanding this complex condition for a more effective management strategy, especially in patients with refractory disease. As we navigate the intricacies of HS, a better patient-centered approach emerges, promising improved outcomes and enhanced quality of life for those grappling with this enigmatic condition.

## Figures and Tables

**Figure 1 reports-07-00013-f001:**
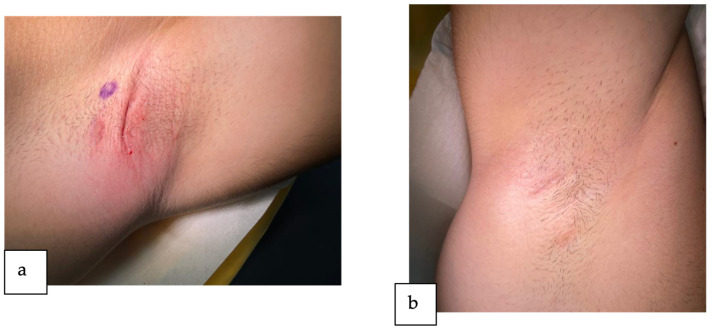
The axillary region of the patient at the current presentation: (**a**) left axilla (**b**) right axilla.

**Figure 2 reports-07-00013-f002:**
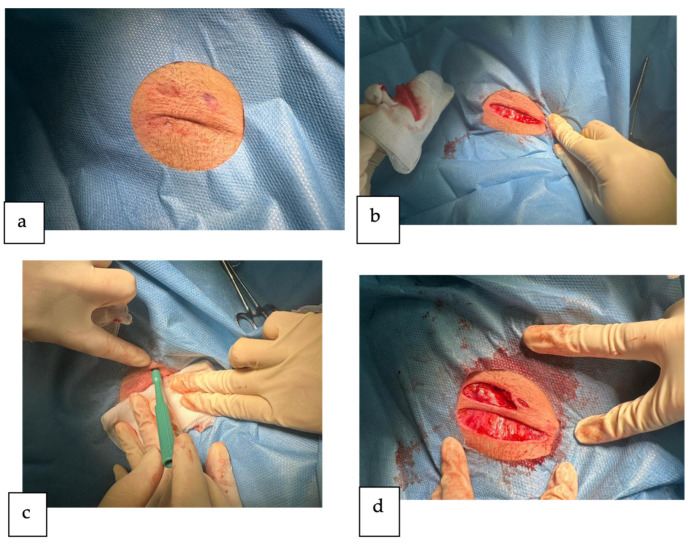
The surgical management of the lesions in the left axilla (**a**) after local anesthesia before the intervention: (**b**) surgical unroofing of one skin tunnel; (**c**) punch debridement of one inflammatory nodule; (**d**) final wounds after surgical unroofing of the second skin tunnel.

**Figure 3 reports-07-00013-f003:**
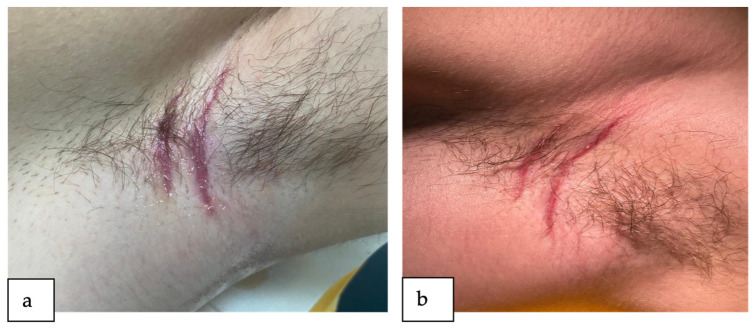
Reevaluation after the surgical treatment of the left axillary lesions (**a**) after 2 weeks post-operation and (**b**) after 4 weeks post-operation.

**Figure 4 reports-07-00013-f004:**
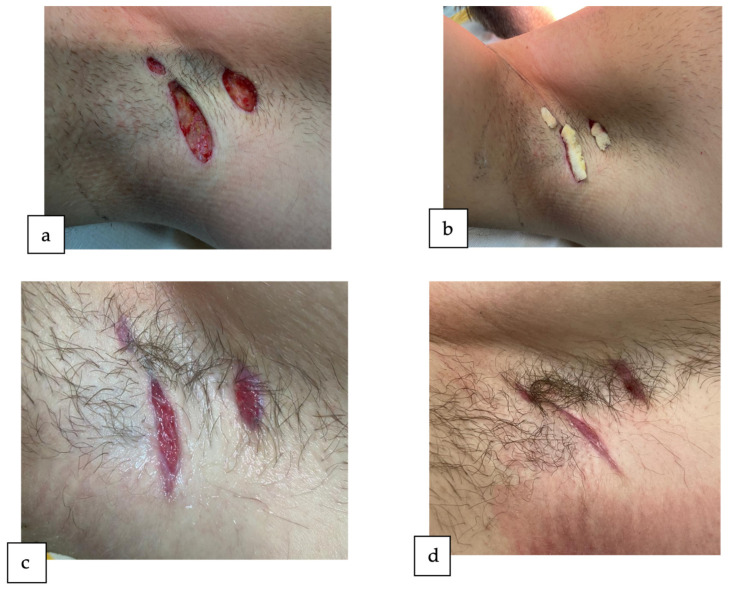
The surgical management of the lesions in the right axilla (**a**) immediately after the surgery; (**b**) wounds left to heal by secondary intention; (**c**) reevaluation after 2 weeks post operation; (**d**) reevaluation after 4-weeks post operation.

## Data Availability

Data is contained within the article.

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
