# Peer review of "Hidradenitis Suppurativa in Association with Ulcerative Proctitis: Surgical Management in a Refractory Case to Topical and Systemic Treatment"

_reports, 2024, doi:10.3390/reports7010013_

Round 1

Reviewer 1 Report

Comments and Suggestions for Authors

The manuscript is interetsting, however it does not inlcude any new information. This is clear that previously the patient was not properly treated and the performed deroofing offered significant clnical outcome. The manuscript may be of intereres only from clinical paoint of view. It is obvious that surgical interevention is required if the tunnels are present.

My comments to the current text:

1. Please descrive IHS4 scoring next to Hurley staging.

2. Please provide more info on the postsyrgical period

3. Please state what is novel in the manuscrpt

Author Response

Please see the attachment below. 

Reviewer 2 Report

Comments and Suggestions for Authors

This paper is a case report and extensive discussion of hidradenitis suppurativa. The authors describe excellent results using simple surgical technique (unroofing). Although there is nothing real or revolutionary in this case report, its value is in the emphasis on single excision and granulation as a solution to hidradenitis tracts. The authors do not describe their unroofing technique, which can vary from surgeon to surgeon. Was a probe used? Was gelatinous material found and simply curetted? Was electrodessication of the base done? The article is fairly well written. Although there are a few minor issues with English phraseology, I would accept it as written and perhaps add a few more details regarding their unroofing technique.

Author Response

Please see the attachment below. 

Round 2

Reviewer 1 Report

Comments and Suggestions for Authors

Adaquately improved manuscript